# On-the-Fly Data Augmentation via Gradient-Guided and Sample-Aware Influence Estimation

## Abstract

Data augmentation has been widely employed to improve the generalization of deep neural networks. Most existing methods apply fixed or random transformations. However, we find that sample difficulty evolves along with the model's generalization capabilities in dynamic training environments. As a result, applying uniform or stochastic augmentations, without accounting for such dynamics, can lead to a mismatch between augmented data and the model's evolving training needs, ultimately degrading training effectiveness. To address this, we introduce SADA, a Sample-Aware Dynamic Augmentation that performs on-the-fly adjustment of augmentation strengths based on each sample's evolving influence on model optimization. Specifically, we estimate each sample's influence by projecting its gradient onto the accumulated model update direction and computing the temporal variance within a local training window. Samples with low variance, indicating stable and consistent influence, are augmented more strongly to emphasize diversity, while unstable samples receive milder transformations to preserve semantic fidelity and stabilize learning. Our method is lightweight, which does not require auxiliary models or policy tuning. It can be seamlessly integrated into existing training pipelines as a plug-and-play module. Experiments across various benchmark datasets and model architectures show consistent improvements of SADA, including +7.3% on fine-grained tasks and +4.3% on long-tailed datasets, highlighting the method's effectiveness and practicality. Code will be made publicly available upon publication.

## 1 Introduction

Data augmentation has been widely adopted for improving the generalization performance of deep neural networks (Yang et al., 2022; Shorten & Khoshgoftaar, 2019; Iglesias et al., 2023). Despite its effectiveness, most existing DA approaches remain static, non-adaptive, and sample-agnostic: they apply either fixed or randomly sampled transformations to all data uniformly, regardless of the evolving difficulty of individual samples or the dynamic learning state of the model in a dynamic training environment (Müller & Hutter, 2021; Cubuk et al., 2019; 2020; Li et al., 2020). For instance, methods such as Cutout (DeVries & Taylor, 2017), AdvMask (Yang et al., 2023), and Mixup (Zhang et al., 2018) generate diverse training data by randomly sampling augmentation parameters. Automatic methods, such as AutoAugment (Cubuk et al., 2019), RandAugment (Cubuk et al., 2020), and DeepAA (Zheng et al., 2022), search for dataset-specific augmentation policy space before training begins and then apply these fixed policies during training. However, this design overlooks a crucial aspect of deep model training: the optimization landscape and the difficulty of individual samples evolving in dynamic training environments. Some samples become easy to fit early on and require increased diversity to avoid redundancy, while others remain hard or unstable and should be preserved in their semantic form to support model refinement. Applying uniform augmentations across these heterogeneous cases introduces a mismatch between augmentation strength and training needs, potentially resulting in noisy updates, degraded sample utility, and suboptimal convergence. Furthermore, many methods often require manual policy tuning or dataset-specific search, which limits scalability across different datasets and architectures (Cubuk et al., 2019; 2020; Yang et al., 2024b). Adaptive augmentation approaches have emerged, but they typically involve bi-level

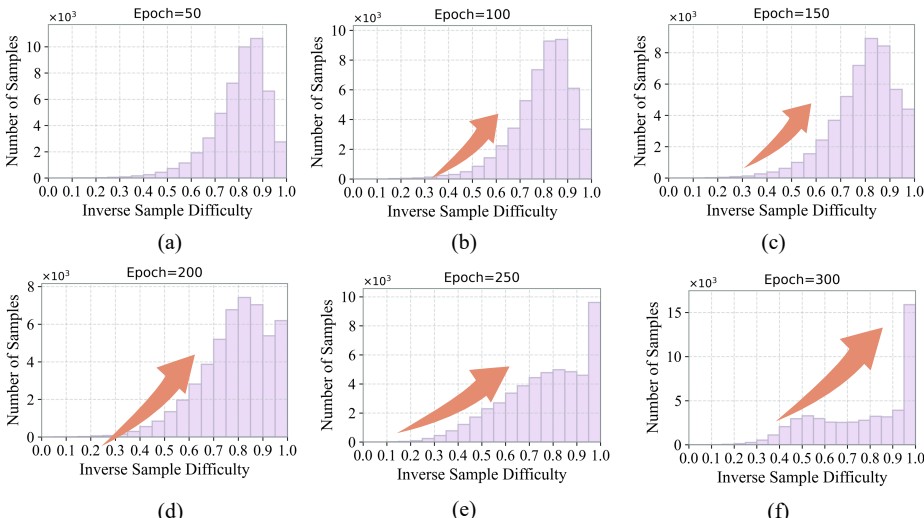

Figure 1: **Evolution of Sample Difficulty Across Training Epochs.** The distribution of sample difficulty evolves dynamically throughout training. A growing proportion of samples becomes easier (higher inverse sample difficulty values), particularly in later epochs. This dynamic trend highlights the necessity of dynamic and sample-aware augmentation strategies during training. Inverse sample difficulty: the reciprocal of sample difficulty.

optimization (Hou et al., 2023), auxiliary models (Suzuki, 2022; Yang et al., 2025), or large search spaces (Bekor et al., 2024), significantly increasing training complexity and resource demand. Thus, a pressing question emerges: *Can we develop an on-the-fly augmentation mechanism that dynamically adapts training data to a model's evolving learning dynamics without sacrificing scalability or efficiency.*

In this paper, we propose SADA, a Sample-Aware Dynamic Augmentation method that performs on-the-fly adjustment of augmentation strength based on each sample's evolving influence during training. Unlike many existing methods that optimize augmentation operations (Bekor et al., 2024; Cubuk et al., 2019), our method uses a unified dataset- and model-agnostic augmentation space (refer to Table 8) and directly modulates augmentation strength. This design offers three benefits: 1). reducing the complexity of the decision space and ensuring efficient online training, 2). providing a more interpretable and fine-grained control over the trade-off between semantic consistency and diversity (Yang et al., 2024a), and 3). eliminating the need for manually crafted or optimization-required dataset-specific augmentation policies and enhancing scalability. To quantify each sample's influence, we project its instantaneous gradient onto the direction of the accumulated model update, thereby capturing how much the sample contributes to the prevailing optimization trajectory. The gradients can be naturally obtained during the standard forward and backward passes, ensuring high efficiency. Furthermore, we compute the temporal variance of this projected influence within a local training window (e.g., 5 epochs), which serves as a proxy for the stability of a sample's learning dynamics. When a sample exhibits consistently low variance, indicating a stable contribution to learning, more substantial augmentation is assigned to promote diversity and avoid overfitting to redundant patterns. Conversely, samples with high variance, suggesting unstable or ambiguous influence, are augmented more conservatively to preserve semantic fidelity and support robust learning. In this way, our method dynamically tailors augmentation magnitudes for each sample based on its training-stage-aware influence. As illustrated in Figure 1, our gradient-guided influence estimation reveals that sample difficulty continuously evolves throughout training: while more samples gradually become easier to fit as the model learns, a small subset remains persistently challenging. By selectively increasing diversity for easier samples and preserving the core semantics of difficult ones, our framework improves generalization while mitigating the risk of introducing ambiguous or disruptive augmentations, highlighting the benefits of our sample-aware, dynamic augmentation.

Experiment results across a variety of benchmark datasets and deep architectures demonstrate consistent and robust performance improvements. On benchmark datasets such as CIFAR-10/100 (Krizhevsky et al., 2009), Tiny-ImageNet (Chrabaszcz et al., 2017), and ImageNet-

1k (Krizhevsky et al., 2017), our approach consistently outperforms existing data augmentation methods. Additionally, we demonstrate strong generalization across different model architectures, including ResNet-based (He et al., 2016) and Vision Transformer (ViT) (Dosovitskiy et al., 2020)-based backbones, etc. Thus, our method can be seamlessly integrated as a plug-and-play component without any modifications to model structures or training schedules. On more challenging long-tailed datasets such as ImageNet-LT and Places-LT (Liu et al., 2019), models trained with our method achieve substantial gains, improving top-1 accuracy by over 4.3% under the closed-set evaluation of ImageNet-LT, highlighting its robustness in imbalanced data scenarios.

Our main contributions can be summarized as follows: **(1)** We propose a lightweight, on-the-fly data augmentation framework that adjusts the augmented data based on the sample-aware evolving influence, without relying on auxiliary models or costly optimization procedures. **(2)** Our method explicitly captures the interplay between data and model by quantifying each sample's contribution to model optimization updates via gradient-guided influence estimation, aligning augmented data with the model's instantaneous learning dynamics. **(3)** Extensive experiments across diverse datasets and architectures demonstrate that our approach serves as a play-and-plug module, consistently improving generalization while maintaining training efficiency.

## 2 RELATED WORK

Data augmentation has long been a fundamental technique for mitigating overfitting and improving the generalization capability of deep neural networks. DA methods have evolved from simple, hand-crafted transformations to more adaptive and automated strategies. It has evolved through multiple methodological paradigms. Early approaches primarily involved applying fundamental transformations, such as rotation, flipping, or cropping (Krizhevsky et al., 2012; Yang et al., 2022), to increase dataset diversity and model robustness. Subsequent advancements focus on developing more sophisticated transformation strategies tailored to specific data characteristics. DA methods can be broadly categorized into image deletion-based, image fusion-based, and automatic policy-based strategies (Müller & Hutter, 2021; Yang et al., 2024b).

**Image Deletion-based Methods.** Cutout (DeVries & Taylor, 2017) introduces regularization by randomly removing square regions from images. GridMask (Chen et al., 2020) generates resolution-matched masks for element-wise multiplication with images. Hide-and-Seek (HaS) (Singh & Lee, 2017) generalizes this idea by partitioning images into grids and stochastically masking subregions. Random Erasing (Zhong et al., 2020) further occludes rectangular areas without resizing. Moreover, AdvMask (Yang et al., 2023) generates learned or structure-aware masking to explicitly target semantic regions, encouraging the model to discover alternative discriminative cues.

**Image Fusion-based Methods.** Fusion-based augmentation synthesizes training samples by blending information across multiple instances. Mixup (Zhang et al., 2018) synthesizes samples via linear interpolation of pixel values and labels across image pairs. However, its indiscriminate blending may produce visually incoherent samples. CutMix (Yun et al., 2019) improves this by replacing rectangular regions between images, preserving spatial structure while introducing inter-sample variability. However, it may still obscure critical semantic content with irrelevant patches. Some improved variants, such as Attentive CutMix (Walawalkar et al., 2020) and PuzzleMix (Kim et al., 2020), incorporate saliency awareness. Despite their effectiveness, these methods typically rely on manually tuned parameters, with limited awareness of the model's evolving training dynamics, potentially limiting the adaptability and optimization efficiency.

**Automated Augmentation Methods.** Automated DA approaches define an augmentation operation space and search for optimal operations and magnitudes. During training, the augmentation operation and corresponding magnitudes are randomly sampled from the pre-defined space. AutoAugment (AA) (Goodfellow et al., 2015) uses reinforcement learning with an RNN controller to predict transformation sequences. Population-Based Augmentation (PBA) (Ho et al., 2019) integrates genetic algorithms with parallel network training, while Fast AutoAugment (Lim et al., 2019) employs Bayesian optimization to discover effective augmentation sequences across partitioned datasets. While powerful, these methods often incur high computational cost and are static once learned. RandAugment (Cubuk et al., 2020) and TrivialAugment (Müller & Hutter, 2021) simplify the parameter spaces through randomized policy selection. EntAugment (Yang et al., 2024b) uses entropy information derived from model snapshots to adjust the augmentation transformations. While effec-

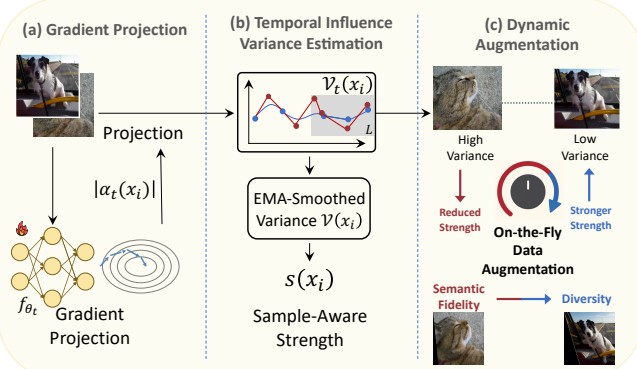

Figure 2: **Overview of Gradient-Guided On-the-Fly Data Augmentation.** At epoch $t$, we quantify the sample's influence on model optimization updates and estimate its stability. The augmentation strength is then adaptively adjusted based on this interplay between model training progress and sample difficulty.

tive, its decisions rely on entropy values extracted from instantaneous model snapshots, which can fluctuate due to the inherent instability of model training. Moreover, ParticleAugment (Tsaregorodtsev & Belagiannis, 2023) proposes a particle filtering scheme for the augmentation policy search. Gradient-based DAS approaches formulate differentiable search spaces, enabling optimization of augmentation strategies. MADAO (Hataya et al., 2020) optimizes models and data augmentation policies simultaneously with Neumann series approximation of the gradients. DADA (Li et al., 2020) formulates data augmentation policy search as a sampling problem and relaxes it into a differentiable framework via Gumbel-Softmax reparameterization. Adversarial variants such as Adversarial AutoAugment (Zhang et al., 2019) and TeachAugment (Suzuki, 2022) generate challenging transformations by maximizing training loss. DDAS (Liu et al., 2021) exploits meta-learning with one-step gradient update and continuous relaxation to the expected training loss for efficient search, without relying on approximations like Gumbel Softmax. In addition, DeepAA (Zheng et al., 2022) progressively constructs multi-layer augmentation pipelines. FreeAugment (Bekor et al., 2024) defines four free degrees of data augmentation and jointly optimizes them. MADAug (Hou et al., 2023), SelectAugment (Lin et al., 2023), SLACK (Marrie et al., 2023), and MetaAugment (Hataya et al., 2022) optimize or learn sample-wise augmentation policies using various techniques, e.g., training an auxiliary policy network. Despite these advances, most existing automated methods overlook the intrinsic heterogeneity of training data difficulty and fail to adapt augmentation intensities dynamically during online training. In contrast, our methodology introduces a lightweight, gradient-based mechanism that samples influence during training and adaptively adjusts augmentation magnitudes in real time, enabling fine-grained, instance-aware data augmentation.

## 3 OUR PROPOSED METHOD

**Overview.** As illustrated in Fig. 2, we propose an on-the-fly data augmentation method that adjusts sample-aware augmentation strength based on each sample's evolving influence on the model's optimization trajectory. Specifically, we project the sample-wise gradient onto the accumulated gradient direction to quantify its contribution to parameter updates. To assess the consistency of this contribution, we compute the variance of the projected values within a local training window and apply EMA smoothing. In this way, the augmentation strengths are dynamically determined in proportion to the stability of the sample's training influence. Samples with low variance, indicating stable influence, are assigned stronger augmentations to improve generalization, while high-variance samples receive milder augmentations to maintain semantic fidelity and stabilize training. In essence, our approach adjusts augmentation strength based on the interaction between the training data and the model's evolving optimization dynamics, thereby achieving dynamic augmentation. During training, we randomly select one augmentation operation from the augmentation space for each sample per epoch and dynamically modulate its strength, which is uniformly applied to various datasets.

Let's denote the whole dataset as $\mathcal{D} = \{(x_i, y_i)\}_{i=1}^N$, where $x_i \in \mathbb{R}^D$, $y_i \in \mathbb{R}^{1 \times K}$, and $K$ is the number of classes. The model $f_\theta$ is trained via gradient descent, updating parameters $\theta$ at step $t$ as:

$$\theta_{t+1} = \theta_t - \eta \sum_{n=1, x_i \in \mathcal{D}}^N g_t(x_i), \tag{1}$$

where $g_t(x_i)$ is the gradient of the loss with respect to sample $(x_i, y_i)$, and $\eta$ is the learning rate.

During training, each sample contributes to the update of the model parameters via its gradient. For samples that are easier to learn, the loss converges rapidly, and their gradient magnitudes tend to stabilize. In contrast, more difficult and ambiguous samples often induce slower loss decay and exhibit persistently fluctuating gradients (Toneva et al., 2019; Zhang et al., 2017; Swayamdipta et al., 2020). To quantify a sample's alignment with the model's current optimization trajectory, we compute the projection of its gradient onto the accumulated update direction. Specifically, we focus on the projection value of the gradient in the direction of parameter updates. The norm of the projected vector is calculated as follows:

$$|\alpha_t(x_i)| = |\langle g_t(x_i), \theta_{t-1} - \theta_t \rangle|. \tag{2}$$

The projected value reflects how much a sample's gradient contributes to the direction of the model's parameter update.

To maintain high efficiency, we approximate the sample-wise gradient projection using first-order Taylor expansion, transforming the gradient-based formulation into a loss-based difference (Zhang et al., 2024). Specifically, the projected influence $\alpha_t(x_i)$ can be approximated as:

$$\begin{aligned} |\alpha_t(x_i)| &= \frac{1}{\eta} \left| (\theta_{t-1} - \theta_t) \nabla_{\theta_{t-1}} \ell\left(f_{\theta_{t-1}}(x_i), y_i\right) \right| \\ &\approx \frac{1}{\eta} \left| \ell\left(f_{\theta_t}(x_i), y_i\right) - \ell\left(f_{\theta_{t-1}}(x_i), y_i\right) \right|, \end{aligned} \tag{3}$$

where $\ell(\cdot)$ denotes the loss function (e.g., cross-entropy). This approximation reduces the need to compute inner products between gradients and parameter updates. In the case of classification tasks with cross-entropy loss, the per-sample loss difference across consecutive steps is given by:

$$\begin{aligned} \Delta \ell_{t-1}^n &= \ell(f_{\theta_t}(x_i), y_i) - \ell(f_{\theta_{t-1}}(x_i), y_i) \\ &= y_i^\top \cdot \log \frac{f_{\theta_t}(x_i)}{f_{\theta_{t-1}}(x_i)}. \end{aligned} \tag{4}$$

To generalize this formulation and enable a fully differentiable approximation, we replace the one-hot label with the soft target $f_{\theta_t}(x_i)^\top$, yielding a KL divergence between the model outputs at two consecutive steps:

$$\Delta \ell_{t-1}^n = f_{\theta_t}(x_i)^\top \cdot \log \frac{f_{\theta_t}(x_i)}{f_{\theta_{t-1}}(x_i)}. \tag{5}$$

This formulation efficiently captures the alignment between a sample's prediction dynamics and model update direction without computing explicit gradients.

To maintain high efficiency during training, we avoid complete historical gradient information and instead approximate sample influence using local training dynamics. Specifically, we compute the variance of sample-wise loss differences over a fixed-size window of the past $L$ epochs, which is:

$$\mathcal{V}_t(x_i) = \sum_{t-L+1}^t \left\| |\Delta \ell_t^n| - \overline{|\Delta \ell_t^n|} \right\|^2, \tag{6}$$

where $\overline{|\Delta \ell_t^n|}$ denotes the average of the absolute loss differences within the window. This formulation provides a local, memory-efficient measure of influence variability and mitigates instability from single-step snapshot assessments. To smooth short-term fluctuations and emphasize recent training dynamics, we update the influence estimate using an exponential moving average:

$$\mathcal{V}(x_i) = \beta \mathcal{V}_t(x_i) + (1 - \beta)\mathcal{V}(x_i), \tag{7}$$

where $\beta$ is the decay coefficient, and both $\beta$ and $L$ are set as constants. In this way, the resulting influence score $\mathcal{V}(x_i)$ shows a proportional relationship with the sample difficulty. To scale the values

Table 1: Image classification accuracy (%) on CIFAR-10/100. * means results reported in the original paper (Müller & Hutter, 2021; Yang et al., 2024b).

| Method | ResNet-44 | ResNet-50 | WRN-28-10 | SS-26-32 | ResNet-44 | ResNet-50 | WRN-28-10 | SS-26-32 |
|---|---|---|---|---|---|---|---|---|
| | CIFAR-10 | | | | CIFAR-100 | | | |
| baseline | $94.10_{\pm.40}$ | $95.66_{\pm.08}$ | $95.52_{\pm.11}$ | $94.90_{\pm.07*}$ | $74.80_{\pm.38*}$ | $77.41_{\pm.27*}$ | $78.96_{\pm.25*}$ | $76.65_{\pm.14*}$ |
| RE | $94.87_{\pm.16*}$ | $95.82_{\pm.17}$ | $96.92_{\pm.09}$ | $96.46_{\pm.13*}$ | $75.71_{\pm.25*}$ | $77.79_{\pm.32}$ | $80.57_{\pm.15}$ | $77.30_{\pm.18}$ |
| RA | $94.38_{\pm.22}$ | $96.25_{\pm.06}$ | $96.94_{\pm.13*}$ | $97.05_{\pm.15}$ | $76.30_{\pm.16}$ | $80.95_{\pm.22}$ | $82.90_{\pm.29*}$ | $80.00_{\pm.29}$ |
| EA | $95.76_{\pm.09}$ | $97.09_{\pm.09}$ | $97.47_{\pm.10}$ | $97.46_{\pm.11}$ | $76.40_{\pm.18}$ | $81.56_{\pm.21}$ | $83.09_{\pm.22}$ | $81.60_{\pm.13}$ |
| TA | $95.00_{\pm.10}$ | $97.13_{\pm.08}$ | $97.18_{\pm.11}$ | $97.30_{\pm.10}$ | $76.57_{\pm.14}$ | $81.34_{\pm.18}$ | $82.75_{\pm.26}$ | $82.14_{\pm.16}$ |
| AA | $95.01_{\pm.11}$ | $96.59_{\pm.04*}$ | $96.99_{\pm.06}$ | $97.30_{\pm.11}$ | $76.36_{\pm.22}$ | $81.34_{\pm.29}$ | $82.21_{\pm.17}$ | $81.19_{\pm.19}$ |
| FAA | $93.80_{\pm.12}$ | $96.69_{\pm.16}$ | $97.30_{\pm.24}$ | $96.42_{\pm.12}$ | $76.04_{\pm.28}$ | $79.08_{\pm.12}$ | $79.95_{\pm.12}$ | $81.39_{\pm.16}$ |
| HaS | $94.97_{\pm.27}$ | $95.60_{\pm.15}$ | $96.94_{\pm.08}$ | $96.89_{\pm.10*}$ | $75.82_{\pm.32}$ | $78.76_{\pm.24}$ | $80.22_{\pm.16}$ | $76.89_{\pm.33}$ |
| DADA | $93.96_{\pm.38}$ | $95.61_{\pm.14}$ | $97.30_{\pm.13*}$ | $97.30_{\pm.14*}$ | $74.37_{\pm.47}$ | $80.25_{\pm.28}$ | $82.50_{\pm.26*}$ | $80.98_{\pm.15}$ |
| Cutout | $94.78_{\pm.35}$ | $95.81_{\pm.17}$ | $96.92_{\pm.09}$ | $96.96_{\pm.09*}$ | $74.84_{\pm.56}$ | $78.62_{\pm.25}$ | $79.84_{\pm.14}$ | $77.37_{\pm.28}$ |
| CutMix | $95.28_{\pm.16}$ | $96.81_{\pm.10*}$ | $96.93_{\pm.10*}$ | $96.47_{\pm.07}$ | $76.09_{\pm.15}$ | $81.24_{\pm.14}$ | $82.67_{\pm.22}$ | $79.57_{\pm.10}$ |
| GridMask | $95.02_{\pm.26}$ | $96.15_{\pm.19}$ | $96.92_{\pm.09}$ | $96.91_{\pm.12}$ | $76.07_{\pm.18}$ | $78.38_{\pm.22}$ | $80.40_{\pm.20}$ | $77.28_{\pm.38}$ |
| AdvMask | $95.49_{\pm.17*}$ | $96.69_{\pm.10*}$ | $97.02_{\pm.05*}$ | $97.03_{\pm.12*}$ | $76.44_{\pm.18*}$ | $78.99_{\pm.31*}$ | $80.70_{\pm.25*}$ | $79.96_{\pm.27*}$ |
| TeachA | $95.05_{\pm.21}$ | $96.40_{\pm.14}$ | $97.50_{\pm.16}$ | $97.29_{\pm.11}$ | $76.18_{\pm.31}$ | $80.54_{\pm.25}$ | $82.81_{\pm.26}$ | $81.30_{\pm.18}$ |
| MADAug | $95.25_{\pm.18}$ | $97.12_{\pm.17}$ | $97.48_{\pm.15}$ | $97.37_{\pm.11}$ | $76.49_{\pm.21}$ | $81.40_{\pm.18}$ | $83.01_{\pm.23}$ | $81.67_{\pm.19}$ |
| SoftAug | $94.51_{\pm.20}$ | $96.99_{\pm.14}$ | $97.15_{\pm.16}$ | $97.22_{\pm.19}$ | $76.41_{\pm.33}$ | $80.94_{\pm.33}$ | $82.61_{\pm.24}$ | $80.33_{\pm.20}$ |
| Ours | $\mathbf{95.87}_{\pm.21}$ | $\mathbf{97.21}_{\pm.10}$ | $\mathbf{97.66}_{\pm.06}$ | $\mathbf{97.51}_{\pm.07}$ | $\mathbf{80.81}_{\pm.41}$ | $\mathbf{81.75}_{\pm.28}$ | $\mathbf{83.17}_{\pm.19}$ | $\mathbf{82.73}_{\pm.15}$ |

of $\mathcal{V}(x_i)$ into the range $[0, 1]$, consistent with the allowable augmentation strength range $m_{max}$, we apply a min-max normalization on it and derive the applied augmentation strengths as $s(x_i) \cdot m_{max}$. When $s(x_i) \to 1$, the augmented samples present a greater variability, and conversely, minor transformations occur as $s(x_i) \to 0$. Importantly, $s(x_i)$ evolves dynamically throughout training, reflecting the model's changing perception of each sample's role in the optimization process. Due to the limited space, we provide the details of the augmentation space and algorithm in Appendix A.

**Theoretical Analysis.** We provide a theoretical analysis to better understand why SADA works. In particular, we show that SADA reduces the empirical Rademacher complexity, thereby tightening the generalization error bound. Formally, the generalization gap is upper-bounded by a term of the form $\mathcal{O}(\frac{1}{n}\sqrt{\sum_i \alpha_i s_i^2})$, where $\alpha_i$ measures sample sensitivity to augmentation and $s_i$ denotes the applied augmentation strength. Optimizing this bound yields a simple allocation rule: augment stable samples more, and unstable samples less. This aligns precisely with the SADA strategy. Therefore, SADA improves generalization from data-centric learning. The complete theoretical derivation is provided in Appendix B.

**Complexity Analysis.** We provide a theoretical analysis showing that SADA introduces negligible computational overhead compared to vanilla training. Specifically, the computational complexity of SADA is $\mathcal{O}(K \times N \times L)$, where $K$ is the total number of classes, $N$ is the number of samples, and $L$ denotes the window length.

## 4 EXPERIMENT

**Datasets and network architectures.** Following prior works (Müller & Hutter, 2021; Yang et al., 2024b; Cubuk et al., 2019), we evaluate our work on a diverse set of benchmark datasets, including CIFAR-10/100 (Krizhevsky et al., 2009), Tiny-ImageNet (Chrabaszcz et al., 2017), and ImageNet-1k (Krizhevsky et al., 2017). To assess its effectiveness in fine-grained recognition tasks, we additionally conduct experiments on Oxford Flowers (Nilsback & Zisserman, 2008), Oxford-IIIT Pets (Parkhi et al., 2012), FGVC-Aircraft (Maji et al., 2013), and Stanford Cars (Krause et al., 2013). Moreover, for evaluating performance under class imbalance, we also conduct experiments using long-tailed datasets, such as ImageNet-LT and Places-LT (Liu et al., 2019). Due to the limited space, more experimental settings are provided in Appendix C.

**Comparison with State-of-the-arts.** We compare our method with a wide range of representative and commonly used methods, including: 1). Cutout (DeVries & Taylor, 2017), 2). HaS (Singh & Lee, 2017), 3). CutMix (Yun et al., 2019), 4) GridMask (Chen et al., 2020), 5). AdvMask (Yang et al., 2023), 6). RandomErasing (Zhong et al., 2020), 7). AutoAugment (AA) (Cubuk et al., 2019), 8). Fast-AutoAugment (FAA) (Lim et al., 2019), 9). RandAugment (RA) (Cubuk et al., 2020), 10). DADA (Li et al., 2020), 11). TeachAugment (TeachA) (Suzuki, 2022), 12). MADAug (Hou et al., 2023), 13). SoftAug (Liu et al., 2023), and 14). TrivialAugment (TA) (Müller & Hutter, 2021).

Table 2: Image classification accuracy (%) on Tiny-ImageNet across various deep models.

| Method | ResNet-18 | ResNet-50 | WRN-50-2 | ResNext-50 |
|---|---|---|---|---|
| baseline | $61.38_{\pm0.99}$ | $73.61_{\pm0.43}$ | $81.55_{\pm1.24}$ | $79.76_{\pm1.89}$ |
| HaS | $63.51_{\pm0.58}$ | $75.32_{\pm0.59}$ | $81.77_{\pm1.16}$ | $80.52_{\pm1.88}$ |
| FAA | $68.15_{\pm0.70}$ | $75.11_{\pm2.70}$ | $82.90_{\pm0.92}$ | $81.04_{\pm1.92}$ |
| DADA | $70.03_{\pm0.10}$ | $78.61_{\pm0.34}$ | $83.03_{\pm0.18}$ | $81.15_{\pm0.34}$ |
| Cutout | $68.67_{\pm1.06}$ | $77.45_{\pm0.42}$ | $82.27_{\pm1.55}$ | $81.16_{\pm0.78}$ |
| CutMix | $64.09_{\pm0.30}$ | $76.41_{\pm0.27}$ | $82.32_{\pm0.46}$ | $81.31_{\pm1.00}$ |
| AdvMask | $65.29_{\pm0.20}$ | $78.84_{\pm0.28}$ | $82.87_{\pm0.55}$ | $81.38_{\pm1.54}$ |
| GridMask | $62.72_{\pm0.91}$ | $77.88_{\pm2.50}$ | $82.25_{\pm1.47}$ | $81.05_{\pm1.33}$ |
| AutoAugment | $67.28_{\pm1.40}$ | $75.29_{\pm2.40}$ | $79.99_{\pm2.20}$ | $81.28_{\pm0.33}$ |
| RandAugment | $65.67_{\pm1.10}$ | $75.87_{\pm1.76}$ | $82.25_{\pm1.02}$ | $80.36_{\pm0.62}$ |
| EntAugment | $70.16_{\pm1.01}$ | $79.06_{\pm1.20}$ | $83.92_{\pm0.97}$ | $81.90_{\pm1.51}$ |
| TeachAugment | $70.05_{\pm0.57}$ | $70.56_{\pm0.44}$ | $82.95_{\pm0.13}$ | $81.39_{\pm0.97}$ |
| TrivialAugment | $69.97_{\pm0.96}$ | $78.41_{\pm0.39}$ | $82.16_{\pm0.32}$ | $80.91_{\pm2.26}$ |
| RandomErasing | $64.00_{\pm0.37}$ | $75.33_{\pm1.58}$ | $81.89_{\pm1.40}$ | $81.52_{\pm1.68}$ |
| Ours | $\mathbf{71.15}_{\pm0.60}$ | $\mathbf{79.66}_{\pm0.52}$ | $\mathbf{84.15}_{\pm0.35}$ | $\mathbf{82.16}_{\pm0.20}$ |

Table 3: Top-1 accuracy (%) on ImageNet-1k dataset with ResNet-50.

| HaS | GM | Cutout | CutMix | Mixup | AA | EA | FAA | RA | MA | SA | DADA | TA | TeachA | Ours |
|---|---|---|---|---|---|---|---|---|---|---|---|---|---|---|
| $77.2_{\pm0.2}$ | $77.9_{\pm0.2}$ | $77.1_{\pm0.3}$ | $77.2_{\pm0.2}$ | $77.0_{\pm0.2}$ | $77.6_{\pm0.2}$ | $78.2_{\pm0.2}$ | $77.6_{\pm0.2}$ | $77.6_{\pm0.2}$ | $\mathbf{78.5}_{\pm0.1}$ | $78.0_{\pm0.1}$ | $77.5_{\pm0.1}$ | $77.9_{\pm0.3}$ | $77.8_{\pm0.2}$ | $78.4_{\pm0.1}$ |

## 4.1 PERFORMANCE COMPARISON

Table 1 compares our method and several widely adopted state-of-the-art baselines on the CIFAR-10 and CIFAR-100 datasets across various deep architectures. While the accuracy margins on these small-scale benchmarks are generally narrow, our method consistently achieves the highest performance across architectures. For example, using WideResNet-28-10 on CIFAR-10, our approach improves accuracy by 2.14% over the best-performing baseline. Similarly, with ResNet-44 on CIFAR-100, we observe a notable performance gain of 7.01%.

To assess scalability, we further evaluate our method on the large-scale Tiny-ImageNet dataset in Table 2. Across different architectures, our method consistently outperforms existing baselines. For instance, on ResNeXt-50, it surpasses the next-best method by over 0.64%, without introducing noticeable training overhead compared to standard training routines. These gains can be attributed to our method's adaptive augmentation mechanism, which dynamically adjusts the augmentation strength based on each sample's influence stability. This design enables a better balance between evolving models and training data, thereby enhancing generalization across models and datasets.

## 4.2 GENERALIZATION ON LARGE-SCALE IMAGENET-1K

We further evaluate the generalization performance of our method on the large-scale ImageNet-1k dataset. Specifically, following experiment settings (Müller & Hutter, 2021), we train ResNet-50 models using different DA methods. As shown in Table 3, our method achieves a competitive performance compared to other baselines. While the accuracy gap between our method and MADAug is marginal, our approach is significantly more efficient, achieving over 2x faster training than MADAug and over 4x faster than TeachAugment, without relying on auxiliary models or bi-level optimization. These results demonstrate that our method offers a compelling trade-off between accuracy and efficiency for large-scale model training.

## 4.3 DATA AUGMENTATION IMPROVES TRANSFER LEARNING

Beyond evaluations on benchmark datasets, we assess model generalization through transfer learning, which tests a model's ability to extract transferable and robust features across domains (Yosinski et al., 2014; Kornblith et al., 2019; Raghu et al., 2019). In this setup, we pretrain ResNet-50 models on CIFAR-100 and Tiny-ImageNet using various data augmentation methods, and then fine-tune them on CIFAR-10.

Table 4: Transferred test accuracy (%) on CIFAR-10 of various DA methods. The pretrained ResNet-50 model is trained on CIFAR-100 (upper row) and Tiny-ImageNet (bottom row).

| baseline | HaS | FAA | DADA | Cutout | CutMix | MADAug | GridMask | AA | EA | RA | TeachAug | TA | RE | Ours |
|---|---|---|---|---|---|---|---|---|---|---|---|---|---|---|
| $91.53_{\pm.03}$ | $92.51_{\pm.24}$ | $92.28_{\pm.13}$ | $92.58_{\pm.09}$ | $92.42_{\pm.20}$ | $92.81_{\pm.47}$ | $92.84_{\pm.10}$ | $91.49_{\pm.10}$ | $92.82_{\pm.04}$ | $92.89_{\pm.19}$ | $92.78_{\pm.23}$ | $92.83_{\pm.18}$ | $92.80_{\pm.16}$ | $92.55_{\pm.05}$ | $\mathbf{93.11}_{\pm.25}$ |
| $64.02_{\pm.05}$ | $66.84_{\pm.06}$ | $70.32_{\pm.63}$ | $69.04_{\pm.43}$ | $65.54_{\pm.75}$ | $69.29_{\pm.09}$ | $72.82_{\pm.32}$ | $64.88_{\pm.43}$ | $69.53_{\pm.53}$ | $72.68_{\pm.73}$ | $64.68_{\pm.23}$ | $69.98_{\pm.17}$ | $71.53_{\pm.35}$ | $64.56_{\pm.27}$ | $\mathbf{77.26}_{\pm.12}$ |

Table 5: Top-1 classification accuracy (%) on ImageNet-LT and Places-LT. * means results reported in the original paper.

| Dataset | Methods | closed-set setting | | | | open-set setting | | | |
|---|---|---|---|---|---|---|---|---|---|
| | | Many-shot | Medium-shot | Few-shot | Overall | Many-shot | Medium-shot | Few-shot | F-measure |
| ImageNet-LT | OLTR | $43.2_{\pm0.1}$* | $35.1_{\pm0.2}$* | $18.5_{\pm0.1}$* | $35.6_{\pm0.1}$* | $41.9_{\pm0.1}$* | $33.9_{\pm0.1}$* | $17.4_{\pm0.2}$* | $44.6_{\pm0.2}$* |
| | OLTR+**Ours** | $\mathbf{46.9}_{\pm0.1}$ | $\mathbf{37.0}_{\pm0.2}$ | $\mathbf{21.6}_{\pm0.2}$ | $\mathbf{36.9}_{\pm0.1}$ | $\mathbf{45.2}_{\pm0.1}$ | $\mathbf{35.6}_{\pm0.2}$ | $\mathbf{20.6}_{\pm0.1}$ | $\mathbf{45.5}_{\pm0.1}$ |
| Places-LT | OLTR | $\mathbf{44.7}_{\pm0.1}$* | $37.0_{\pm0.2}$* | $25.3_{\pm0.1}$* | $35.9_{\pm0.1}$* | $\mathbf{44.6}_{\pm0.1}$* | $36.8_{\pm0.1}$* | $25.2_{\pm0.2}$* | $46.4_{\pm0.1}$* |
| | OLTR+**Ours** | $44.3_{\pm0.1}$ | $\mathbf{40.8}_{\pm0.2}$ | $\mathbf{28.9}_{\pm0.2}$ | $\mathbf{38.5}_{\pm0.1}$ | $44.1_{\pm0.1}$ | $\mathbf{40.6}_{\pm0.2}$ | $\mathbf{28.6}_{\pm0.1}$ | $\mathbf{50.4}_{\pm0.2}$ |

This evaluation is motivated by the fact that stronger data augmentation strategies can lead to more generalizable feature representations. As shown in Table 4, it can be observed that our method achieves consistently higher accuracy after transfer compared to baseline augmentation approaches, regardless of the pertaining dataset. These results indicate that models trained with our dynamic augmentation strategy learn more transferable and semantically meaningful features, further validating the generalization benefits of our approach.

### 4.4 RESULTS ON FINE-GRAINED DATASETS

To further assess the versatility of our method, we evaluate its performance on several fine-grained classification benchmarks, including Oxford Flowers (Nilsback & Zisserman, 2008), Oxford-IIIT Pets (Parkhi et al., 2012), FGVC-Aircraft (Maji et al., 2013), and Stanford Cars (Krause et al., 2013). These datasets are characterized by subtle inter-class differences, making them particularly challenging for standard data augmentation strategies.

As shown in Table 6, incorporating our method into the standard training process can significantly enhance model performance. Notably, on the Oxford Flower dataset, it achieves over 8% absolute improvement compared to baseline learning. These results highlight the effectiveness of our sample-aware augmentation approach in fine-grained scenarios.

### 4.5 RESULTS ON LONG-TAILED DATASETS

While most existing DA methods are not evaluated on long-tailed datasets, we further evaluate the robustness of our method on more challenging long-tailed benchmarks, i.e., ImageNet-LT and Places-LT (Liu et al., 2019), which exhibit significant class imbalance. We closely follow the experimental setting in OLTR (Liu et al., 2019), using the same network backbone and evaluation metrics, except utilizing our augmentation method. As shown in Table 5, our method achieves consistent performance improvements across both closed-set and open-set evaluation settings. On ImageNet-LT, we improve the overall top-1 accuracy by 1.3% in the closed-set scenario. On Places-LT, our method increases the F-measure by 4% in the open-set setting. These results highlight the ability of our adaptive augmentation strategy to improve generalization under severe data imbalance, without requiring explicit rebalancing techniques or auxiliary supervision.

### 4.6 CROSS-ARCHITECTURE GENERALIZATION

In Table 1 and Table 2, we demonstrate the effectiveness of our method across various CNN-based architectures. To further evaluate its generalizability, we extend our experiments to Vision Transformer-based models using the ImageNet-1k dataset. As shown in Table 7, our method yields consistent performance gains for both ViT variants, improving the performance of ViT-Base/Large/Huge on ImageNet-1k. Importantly, these gains are achieved without introducing large additional training overheads, highlighting the efficiency of our method. Consequently, these results confirm that our method is architecture-agnostic and can be seamlessly integrated into training pipelines as a plug-and-play module to improve performance.

Table 6: Test accuracy (%) on fine-grained datasets with ResNet-50.

| Dataset | baseline | Ours |
|---|---|---|
| Oxford Flowers | 89.47$_{\pm 0.08}$ | **98.04**$_{\pm 0.09}$ |
| Oxford-IIIT Pets | 89.73$_{\pm 0.18}$ | **92.53**$_{\pm 0.12}$ |
| FGVC-Aircraft | 77.25$_{\pm 0.09}$ | **80.76**$_{\pm 0.12}$ |
| Stanford Cars | 82.13$_{\pm 0.03}$ | **91.89**$_{\pm 0.07}$ |

Table 7: Test accuracy (%) on ImageNet-1k with ViT-Base/Large/Huge.

| Model | baseline | Ours |
|---|---|---|
| ViT-B | 82.30 | **83.38**$_{\uparrow 1.08}$ |
| ViT-L | 84.47 | **85.01**$_{\uparrow 0.54}$ |
| ViT-H | 85.91 | **86.88**$_{\uparrow 0.97}$ |

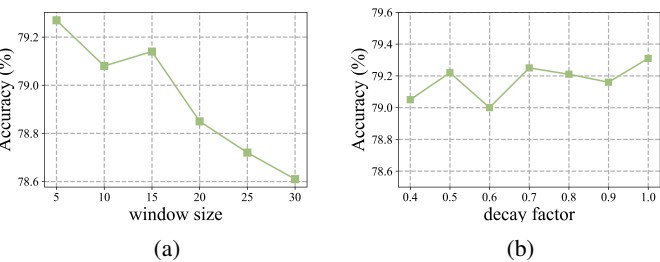

(a)  (b)

Figure 3: The stability of our method on the two parameters, i.e., the window size and the decay factor, with CIFAR-100 using ResNet-18.

## 4.7 EFFICIENCY COMPARISON

We compare the training costs of our method with other baselines. As illustrated in Figure 4, in the efficiency-effectiveness plane, our method achieves a favorable trade-off between training cost and performance. Consistent with the complexity analyses in Section 3, our approach introduces negligible additional overhead compared to standard training. This is primarily because the required gradient information can be directly obtained during standard forward and backward passes, without relying on auxiliary networks or a complex optimization process. While our method incurs slightly

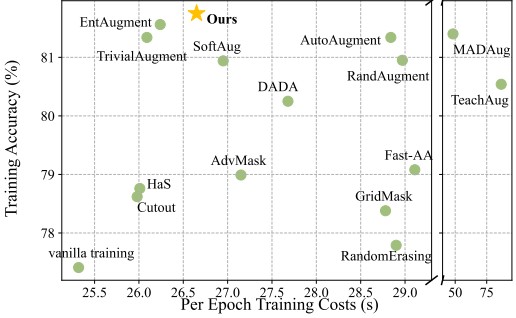

Figure 4: Comparison in the effectiveness-efficiency tradeoff. We report the average per-epoch training costs using a 2-NVIDIA-RTX2080TI-GPUs server.

higher training costs than baselines such as Cutout, HaS, and TrivialAugment, the difference is minimal. Importantly, our method consistently delivers better performance, achieving a better balance between efficiency and accuracy.

## 4.8 ABLATION STUDY

We conduct an ablation study to investigate the effect of two hyperparameters in our method: the window size $L$ in Eq. equation 6 and decay factor in Eq. equation 7. As shown in Figure 3(a), increasing the window size $L$ leads to a consistent drop in classification accuracy. This is because larger windows oversmooth the instantaneous dynamics of sample influence, thereby delaying the dynamic augmentation's responsiveness to model training dynamics. As a result, maintaining a small window size not only better captures the local importance of each sample but also reduces the memory costs. Figure 3(b) shows the effect of varying the decay factor $\beta$. The model performance remains generally stable across different $\beta$ values, indicating that our method is robust to it.

## 5 CONCLUSION

This paper proposes a novel on-the-fly data augmentation method that performs sample-aware augmentation by modeling the evolving interplay between data and the model during training. Unlike existing approaches, our proposed method leverages a dynamic augmentation mechanism, mitigating overfitting for stable samples by increasing their diversity while promoting generalization for uncertain ones by preserving semantic fidelity. We hope our work inspires further research on train-dynamic-aware data augmentation from an on-the-fly perspective and believe our method will serve as a promising plug-and-play tool for the community, enabling enhanced deep model training.

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

## A    MORE DETAILS OF THE METHOD

Table 8: Our employed augmentation operations with corresponding magnitude ranges across different datasets (Müller & Hutter, 2021; Yang et al., 2024b), only including lightweight image transformations.

| Transformation | Max allowable magnitude |
|---|---|
| identity | - |
| auto contrast | - |
| equalize | - |
| color | $+1.9$ |
| contrast | $+1.9$ |
| brightness | $+1.9$ |
| sharpness | $+1.9$ |
| rotation | $\pm30°$ |
| translate$_x$ | $\pm10$ |
| translate$_y$ | $\pm10$ |
| shear$_x$ | $\pm0.3$ |
| shear$_y$ | $\pm0.3$ |
| solarize | $+256$ |
| posterize | $+4$ |

**Algorithm: Detailed algorithm pipeline of our method**

**Require:** Training dataset $\mathcal{D}$, network $f_\theta$ with weights $\theta$, decay coefficient $\beta$
1: **for** each training step $t = 0, 1, \ldots$ **do**
2:     Sample a mini-batch $\{\boldsymbol{x}_i, y_i\}_{i=1}^B$ from $\mathcal{D}$
3:     Compute predicted probabilities $f_\theta(x_i)$ for each $\boldsymbol{x}_i$
4:     Update model weights according to Eq. 1
5:     Compute $\Delta\ell_t^n$ for each $\boldsymbol{x}_i$ (Eq. 5)
6:     Compute $\mathcal{V}_t(x_i)$ in one window (Eq. 6)
7:     Update $\mathcal{V}_t(x_i)$ with EMA (Eq. 7)
8:     Compute $s(x_i)$
9:     Augment samples with $s(x_i)$ in next epoch
10: **end for**

## B    THEORETICAL JUSTIFICATION

We provide a sketch of theoretical justification showing why our sample-adaptive augmentation (SADA) strategy—assigning stronger augmentation to stable (low-variance) samples and weaker augmentation to unstable (high-variance) samples—can improve generalization.

**Setup.**    Let $\mathcal{D} = \{(x_i, y_i)\}_{i=1}^n$ be the training set, with feature map $\phi : \mathcal{X} \to \mathbb{R}^d$ satisfying $\|\phi(x)\| \leq R$. The hypothesis class is $f_\theta(x) = \langle\theta, \phi(x)\rangle$ with $\|\theta\| \leq B$. For each sample, an augmentation operator $\mathcal{A}_{s_i}$ with magnitude $s_i \in [0, s_{\max}]$ generates

$$\phi(\tilde{x}_i) = \mu_i(s_i) + \Delta_i(s_i), \quad \mathbb{E}[\Delta_i(s_i) \mid x_i] = 0.$$

We assume (i) the loss $\ell$ is $L_\ell$-Lipschitz in its prediction, and (ii) the fluctuation satisfies $\mathbb{E}\|\Delta_i(s_i)\|^2 \leq \alpha_i s_i^2$, where $\alpha_i$ encodes the sample's sensitivity to augmentation.

**Rademacher complexity with augmentation.**    The empirical Rademacher complexity of the augmented class is

$$\mathfrak{R}_n(\mathcal{F}_s) = \frac{1}{n} \mathbb{E}_{\varepsilon,\tilde{x}}\Big[ \sup_{\|\theta\|\leq B} \sum_{i=1}^n \varepsilon_i \langle\theta, \phi(\tilde{x}_i)\rangle \Big].$$

Decomposing into mean and fluctuation terms and applying Khintchine–Kahane inequality yields

$$\mathfrak{R}_n(\mathcal{F}_s) \leq \frac{B}{n}\Big( \sqrt{\sum_{i=1}^n \|\mu_i(s_i)\|^2} + \sqrt{\sum_{i=1}^n \alpha_i s_i^2} \Big).$$

By the contraction lemma, the loss class satisfies

$$\mathfrak{R}_n(\ell \circ \mathcal{F}_s) \leq \frac{L_\ell B}{n}\Big( \sqrt{\sum_{i=1}^n \|\mu_i(s_i)\|^2} + \sqrt{\sum_{i=1}^n \alpha_i s_i^2} \Big).$$

Thus, the generalization gap is controlled by a complexity term $\frac{L_\ell B}{n}\sqrt{\sum_i \alpha_i s_i^2}$.

**Optimal allocation.** Suppose we require $\sum_i s_i \geq S$. Minimizing $\sum_i \alpha_i s_i^2$ under this constraint gives the strictly convex problem

$$\min_{0 \leq s_i \leq s_{\max}} \sum_{i=1}^n \alpha_i s_i^2 \quad \text{s.t.} \quad \sum_{i=1}^n s_i \geq S.$$

The KKT conditions yield a water-filling solution:

$$s_i^* = \min\left\{ s_{\max}, \frac{\lambda}{2\alpha_i} \right\}, \quad \sum_i s_i^* = S.$$

Therefore, the optimal strategy assigns *larger augmentation strength to samples with smaller $\alpha_i$* (i.e., lower sensitivity), and smaller strength to those with larger $\alpha_i$ (higher sensitivity).

**Connection to variance measure.** In our method, $\alpha_i$ is bounded by a constant multiple of the variance measure $\mathcal{V}(x_i)$ computed from the gradient dynamics, i.e., $\alpha_i \leq c\mathcal{V}(x_i)$. Hence, the optimal allocation $s_i^*$ is monotone decreasing in $\mathcal{V}(x_i)$, which aligns exactly with our SADA rule: *low-variance samples receive stronger augmentation, while high-variance samples receive weaker augmentation.*

## C  IMPLEMENTATION DETAILS

Our experiments are conducted across a wide range of network architectures, including ResNet-based models, e.g., ResNet-18/50 and Wide ResNet, ViT-based models, e.g., ViT-Base/Large/Huge, and architectures with advanced regularization such as Shake-Shake (Gastaldi, 2017) and ResNeXt (Xie et al., 2017). This setup allows us to comprehensively evaluate the generalization and scalability of our method across different data domains and architectural families. Some results for baseline methods are taken from the original publications Yang et al. (2024b); Müller & Hutter (2021); Cubuk et al. (2019).

Our experimental setup follows standard practices established in prior works (DeVries & Taylor, 2017; Yang et al., 2023; 2024b; Chen et al., 2020; Müller & Hutter, 2021). Specifically, during online training, only augmented data is used for model optimization, without incorporating original data. Unless otherwise specified, we train all models for 300 epochs using a batch size of 256, an initial learning rate of 0.1, SGD with momentum 0.9, weight decay of $5e-4$, and a cosine annealing learning rate decay strategy. Input images undergo standard preprocessing with random cropping and horizontal flipping, consistent with the augmentation setup used for the baseline methods. For experiments involving the Shake-Shake model, we follow the established protocol (Gastaldi, 2017) and train for 1800 epochs using SGD with Nesterov Momentum, weight decay of $1e-3$, and cosine learning rate decay. The augmentation operation space used is consistent with prior works (Müller & Hutter, 2021; Yang et al., 2024b). Unless otherwise stated, we use ResNet-50 as the default architecture. We consistently set the window size as 10 and the decay factor as 0.9 across all the tasks and datasets without any dataset- or architecture-specific tuning. The consistent improvements across settings demonstrate that SADA is robust and not sensitive to these hyperparameters in practice. For all experiments, we report the average and standard deviation of test accuracy over three independent runs. Note that because of the huge calculation consumption on ImageNet-1k, the experiment in each case is performed once.

## D  PERFORMANCE UNDER CONTROLLED RANDOMNESS OF THE AUGMENTATION OPERATIONS

Table 9: Performance under different numbers of augmentation operations in our augmentation space on CIFAR-100 using ResNet-50.

| # of operations | 4 | 6 | 8 | 10 | 12 | 14 |
|---|---|---|---|---|---|---|
| Acc. (%) | 81.6 | 81.5 | 81.6 | 81.8 | 81.9 | 81.8 |

In this section, we evaluate the performance of our method under different controlled randomness. As shown in Table 9, it can be observed that reduced randomness in augmentation operations brings

minimal influence on our method. SADA remains highly stable across different levels of operation randomness. Therefore, we validate that the superior effectiveness of SADA stems from the adaptive adjustment of augmentation strengths, rather than from the random selection of operations.

## E  TRAINING COSTS ANALYSIS

Table 10: Wall-clock time (h) of baseline vs. SADA on ImageNet-1k using a 4-A100-GPU server.

|  | ResNet-50 | ViT-B | ViT-L |
|---|---|---|---|
| Baseline | 22.1 | 149.1 | 363.2 |
| SADA | 22.5 | 150.8 | 366.4 |
| Increased costs | +1.8% | +1.1% | 0.8% |

In this section, we further analyze SADA's actual training costs. As shown in Table 10, it can be observed that SADA incurs no noticeable additional training cost compared to standard training. This is because we adopt a first-order Taylor expansion to convert the gradient-projection term into a loss-difference formulation (Eq. 6), which can be obtained directly from the forward pass. This avoids any additional gradient calculation beyond standard training, and thus the resulting computational overhead introduced by SADA is minimal.

## F  COMPARISON WITH ENTAUGMENT

Recently, adaptive data augmentation methods have shown strong effectiveness, and both SADA and EntAugment fall within this broader family of approaches that adjust augmentation strength based on per-sample behavior during training. While we empirically compare SADA and EntAugment across various evaluation settings, here we outline their methodological differences to provide a clearer understanding. 1). Different signals. EntAugment uses classification entropy from model snapshots, while SADA instead uses gradient-based influence projection to measure how each sample directly contributes to the optimization trajectory. 2). Different stability mechanisms. EntAugment can fluctuate across training, while SADA incorporates the temporal variance of sample influence over a local window, providing a more stable and reliable indicator of the learning effect. 3). Different training-stage awareness: EntAugment's entropy does not explicitly capture how a sample's effect evolves over time. SADA naturally reflects evolving sample dynamics via gradient influence and its temporal consistency. 4). Different optimization basis. EntAugment relies on a heuristic uncertainty signal. SADA is grounded in optimization theory, using gradients and accumulated updates to modulate augmentation in a way that is directly aligned with the learning process.

In summary, while the two methods share similarities, SADA adopts a fundamentally different mechanism that is more stable, more training-aware, and more closely aligned with underlying optimization dynamics. Thus, while EntAugment provides promising performance, SADA achieves stronger effectiveness.

## G  MORE COMPARISON WITH THE PUBLISHED RESULTS OF TRIVIALAUGMENT

Table 11: Comparison with the published results of TrivialAugment (TA) using the experimental setting from Müller & Hutter (2021) on CIFAR-10/100.

| Dataset | Method | Baseline | TA | Ours |
|---|---|---|---|---|
| CIFAR-10 | WRN-28-10 | 97.0 | 97.5 | **97.9** |
|  | SS-26-96 | 97.5 | 98.2 | **98.4** |
| CIFAR-100 | WRN-28-10 | 82.2 | 84.3 | **84.6** |
|  | SS-26-96 | 83.3 | 86.2 | **86.7** |

In addition to the comparisons with TrivialAugment (TA) in Section 4, in this section, we compare with TA's published results using its training configurations. As shown in Table 11, under the identical settings, SADA consistently surpasses TA across deep models and datasets.

## H    SOCIETAL IMPACT STATEMENT

This work focuses on improving the generalization and training efficiency of deep learning models through a sample-aware data augmentation framework, SADA. The potential positive societal impacts include reducing the reliance on large-scale, manually curated datasets by enabling more effective use of limited or imbalanced data, which can lower data collection costs and broaden access to machine learning in resource-constrained settings. In particular, the method's plug-and-play nature and computational efficiency may benefit applications in healthcare, environmental monitoring, or education, where robust generalization under limited data is critical.

## I    DISCUSSION AND FUTURE WORK

In this section, we discuss some potential limitations and future work for our method.

Since our method computes the variance of gradient-based influence signals to determine sample-wise augmentation strengths, it requires maintaining a local history of these values within a sliding window and introduces two parameters: window size $L$ and decay factor $\beta$. In all our experiments across datasets and architectures, we adopt the same default hyperparameter configuration ($L = 10$ and $\beta = 0.9$) without any dataset-specific or model-specific tuning. To ensure the responsiveness of augmentation strength to recent training dynamics, our framework favors small window sizes, thus capturing meaningful local variations. Meanwhile, our ablation studies confirm that the decay factor is highly stable. These findings suggest that our framework is robust to hyperparameter choices. To provide clearer parameter setting suggestions in practice, based on our ablation study results, we summarize these insights: using $L = 5, 10$ with $\beta = 0.9$, without large-scale tuning.

Currently, our method is designed and evaluated primarily for supervised image classification tasks. While the sample-aware augmentation principle is general, its application to other domains, such as object detection, semantic segmentation, or image generation, remains underexplored. These tasks involve fundamentally different training objectives and model behaviors, and investigating how gradient-guided influence estimation interacts with task-specific objectives and model architectures will be an important direction for future work.

## J    AI ASSISTANT USAGE STATEMENT

During the preparation of this paper, we made only moderate use of large language models for text polishing.

## K    REPRODUCIBILITY

Implementation will be made publicly available.

