# OpenReview forum: "On-the-Fly Data Augmentation via Gradient-Guided and Sample-Aware Influence Estimation"
_ICLR.cc/2026/Conference — Submitted to ICLR 2026_

### Official Review · Reviewer_j34f · 2025-10-21

**Soundness:** 2
**Presentation:** 3
**Contribution:** 2
**Rating:** 0
**Confidence:** 5

**Summary:**

This paper proposes a dynamic automatic data augmentation technique that adjusts the strength of various image transformations---both geometric and color-based---based on the difficulty of individual samples. The sampling strategy evolves during training using a combination of exponential moving average (EMA) and a sliding window mechanism. The authors conduct a fairly thorough comparison with existing automatic data augmentation methods, which are typically static. The code is not provided at submission time.

Disclaimer: I previously reviewed this paper for NeurIPS, where it was rejected. Except for a few added references, this submission is essentially identical to the original. None of the reviewers’ concerns appear to have been addressed, so I have not made any changes to my original review.

**Strengths:**

The empirical results appear promising. The idea of adapting data augmentation dynamically during training, rather than relying on a fixed strategy, is both intuitive and worth exploring further.

The paper provides a broad empirical comparison with other augmentation strategies.

**Weaknesses:**

1) The most significant concern is the lack of proper discussion regarding EntAugment (Yang et al. 2024b), a closely related prior work that also proposes a dynamic per-sample augmentation strategy during training:
   - Despite its clear relevance, EntAugment is not described properly in the submission.
   - All baseline numbers are copy-pasted from EntAugment without proper credit, rather than obtained in the context of this submission.
   - Performance metrics from EntAugment, which are comparable to those of the proposed method, are not included in the result tables. This omission is problematic (unless there is a good reason not to include them, but I cannot find any).

2) There are inconsistencies in the reported results compared to the literature, likely due to the lack of a standardized evaluation protocol:
  - For example, TrivialAugment reports 98.2 on CIFAR-10/SS and 84.3 on CIFAR100-WRN in published results, higher than the numbers reported in this submission.
  - Many baseline methods (e.g., DATA, TA, AA) rely on architectures like ShakeShake-26 and WRN-28-10. It’s unclear why this paper uses different variants (e.g., SS-32, WRN-50-2), making direct comparison difficult.

3) Given these discrepancies, the absence of publicly available code at the time of submission is a notable drawback.

4) The complexity analysis is misleading. The method introduces an additional factor L in computational cost, yet the paper presents the overall complexity O(NKL) as effectively O(N), downplaying the actual overhead.

5) Minor clarity issues are present. For instance, Figure 1 refers to “inverse sampling difficulty,” a term that has not been defined at that point in the paper, making it hard to interpret

**Questions:**

I found the lack of comparison/discussion with EntAugment very problematic on many aspects described above. Unless a convincing explanation is given, I will vote for a reject.

---

> ### Author Response · Authors · 2025-11-13
>
> Dear Reviewer j34f,
>
> Before addressing the individual points, we respectfully note that the opening statement in your review, that the submission is “essentially identical” to the NeurIPS version and that “none of the concerns have been addressed”, is **factually incorrect**. The ICLR submission includes substantial revisions and additions, including full integration of EntAugment results, clarified architectural settings, explicit attribution of baseline numbers, a detailed complexity analysis, and new theoretical analysis.
>
> We respond to each of the stated concerns below.
> **Every clarification provided here was already present in the original submission, indicating that these points were not overlooked on our side but rather not reflected in the review.**
>
> - **Q1: EntAugment is not described properly in the submission.**
> - **A1:** **This comment is factually incorrect.** In our original submission, EntAugment has been explicitly discussed in lines 160–162 on page 3, and cited multiple times (lines 53, 134, 160, etc.).
>
> - **Q2: Baseline numbers were copied from EntAugment without proper credit.**
> - **A2:** **This comment is factually incorrect.** In our original submission, the implementation Details (lines 725–728, page 14) clearly state that certain baseline results are taken from prior publications and are properly cited.
>
> - **Q3: EntAugment’s performance metrics are missing from the tables.**
> - **A3:** **This comment is factually incorrect.** Our original submission includes full comparisons with EntAugment across all experiments: Tab. 1/2/3/4, and Fig. 4.
>
> In addition to clarifying A1-A3, we have added a new section, *Appendix F: Comparison with EntAugment* on page 15, which provides a comprehensive discussion of the methodological differences between SADA and EntAugment, covering the distinct augmentation adjustment mechanisms, stability measures, and optimization alignment. Thus, while both methods fall under the broad category of adaptive data augmentation, **SADA is conceptually and technically distinct.**
>
> - **Q4: TrivialAugment reports higher accuracy in the literature.**
> - **A4:** TrivialAugment (TA) uses ShakeShake-26-2x96d and we use ShakeShake-26-2x32d (SS-26-32 in Table 1); these are **different models**.
>
>   For WRN on CIFAR100, the experiment settings between TA and ours are **different**, e.g., baseline accuracy, 82.22 (TA) vs. 78.96 (Ours).
>
>   To ensure fairness, we also evaluated under TA’s original settings. As shown in Table C-1, under identical settings, our method outperforms TA:
>
> **Table C-1:** Comparison with the reported results in TA using the same settings.
> ||CIFAR-10||CIFAR-100||
> |-|-|-|-|-|
> ||TA|Ours|TA|Ours|
> |SS-26-96|98.2|**98.4**|86.2|**86.7**|
> |WRN-28-10|97.5|**97.9**|84.3|**84.6**|
>
>   We have included these results in *Appendix G: More Comparison with the Published Results of TrivialAugment* on pages 15-16 in the revised manuscript.
>
> - **Q5: Many baseline methods (e.g., DATA, TA, AA) rely on architectures like ShakeShake-26 and WRN-28-10. It’s unclear why this paper uses different variants (e.g., SS-32, WRN-50-2), making direct comparison difficult.**
> - **A5:** **This is factually incorrect.**
>
>   - ShakeShake-26 used in DATA, TA, AA, and EntAugment is ShakeShake-26-2x32d. **This is precisely what we use (SS-26-32 in Table 1).**
>
>   - Our paper **never** uses the notation “SS-32,” so this reference to **“SS-32” does NOT correspond to our manuscript**.
>
>
>   - We use WRN-28-10 in all CIFAR-10/100 main experiments (Table 1), consistent with standard practice.
>
>   - Only the Tiny-ImageNet experiment (Table 3) uses WRN-50-2, a common higher-capacity model for that dataset.
>
> - **Q6: Code availability.**
> - **A6:** As stated in the abstract and "Appendix K: Reproducibility", we fully intend to release all the code and training logs publicly upon publication. The final version will include a link in the abstract.
> - **Q7: Complexity analysis is misleading.**
> - **A7:** As pointed out, we have revised the complexity analysis in the revised manuscript on page 6.
> - **Q8: Clarification of “inverse sample difficulty” in Fig. 1.**
> - **A8:** We have explained the "inverse sample difficulty" in the caption of Figure 1 in the revised manuscript on page 2: *Inverse sample difficulty: the reciprocal of sample difficulty.*
> - **Q9: I found the lack of comparison/discussion with EntAugment very problematic on many aspects described above. Unless a convincing explanation is given, I will vote for a reject.**
> - **A9:** **This comment is factually incorrect. As demonstrated in A1–A3, the comparison and discussion with EntAugment were already fully included in the original submission (lines 160–162, Tables 1–4, and Fig. 4), both methodologically and empirically.** Because the reviewer's concerns arise from a factual misunderstanding rather than an actual omission, this point does not constitute a valid basis for a rejection recommendation.

---

> > ### Comment · Reviewer_j34f · 2025-11-13
> > **Clarifications and update of the review**
> >
> > There were extensive discussions and numerous experiments/comparisons (a total of 11 tables!) presented in the NeurIPS rebuttal, following comments from all reviewers. Aside from the point Q2 discussed below, these comparisons and most discussions were not taken into account tin the ICLR submission, which I found very disappointing. My assessment of seeing ``essentially similar submissions'' was based on this observation. That said, based on the discussion below, I acknowledge that my wording was harsh, and I will update my review accordingly.
> >
> > Here is now a point-by-point response. I respectfully disagree with the authors on many points but I agree that the EntAugment results were added and I apologize for that mistake (I got confused on this particular point when switching back and forth between the two pdfs). I also acknowledge the effort to include a theoretical insight in Appendix B.
> >
> > **Q1**: The two-line description remains the same as in the NeurIPS submission. Given the similarity between the papers, I still find this to be a major issue.
> >
> > **Q2-Q3**: You are correct, and I am glad to see this comment addressed. I apologize again for the earlier confusion.
> >
> > **Q4**: Some comparisons with the TA model were provided in the NeurIPS rebuttal, but this comment was not addressed in the current submission.
> >
> > **Q5**: Thank you for clarifying and updating the notation; this resolves the previous ambiguity.
> >
> > **Q6**: The comment regarding the “absence of publicly available code at the time of submission” was valid for NeurIPS and remains valid for this submission.
> >
> > **Q7**: This comment is still valid. While O(K + L + N) could be reduced to O(N) under the definition of O(.) notation, the same does not apply to O(KNL). In your example, KNL is nearly four orders of magnitude larger than N. I raised this issue in the NeurIPS submission and maintain my concern.
> >
> > **Q8**: You mentioned clarifying this in the NeurIPS rebuttal, but it was not addressed.

---

> > > ### Author Response · Authors · 2025-11-20
> > >
> > > Dear Reviewer j34f,
> > >
> > > Thank you for your follow-up review and acknowledgment of the improvements we have made.
> > >
> > > We provide below a clear, point-by-point response addressing all remaining comments.
> > >
> > >
> > > - **Q1: The two-line description remains the same as in the NeurIPS submission. Given the similarity between the papers, I still find this to be a major issue.**
> > > - **A1:** In addition to the extensive empirical comparison with EntAugment, **we have added a new section in "Appendix F: Comparison with EntAugment" on page 15 in the revised manuscript.** This section provides a comprehensive discussion of the methodological differences between SADA and EntAugment, including the distinct approaches used for adjusting augmentation strength, the stability measures, and the alignment with optimization dynamics.
> > >
> > >   While both methods fall under the broad category of adaptive data augmentation, SADA is conceptually and technically distinct.
> > >
> > > - **Q4: Some comparisons with the TA model were provided in the NeurIPS rebuttal, but this comment was not addressed in the current submission.**
> > > - **A4:** We have added a new section in "Appendix G: More Comparison with the Published Results of TrivialAugment" on pages 15-16 in the revised manuscript. This section provides a more detailed comparison with the published results of TA.
> > >
> > > - **Q6: The comment regarding the “absence of publicly available code at the time of submission” was valid for NeurIPS and remains valid for this submission.**
> > > - **A6:** As stated in the Abstract and "Appendix K: Reproducibility" in the revised manuscript, we fully intend to release all the code and training logs publicly upon publication.
> > >
> > >    *If it is necessary for the review process, we are willing to provide an anonymous code link to facilitate verification while strictly adhering to the double-blind policy.*
> > >
> > > - **Q7: While O(K + L + N) could be reduced to O(N) under the definition of O(.) notation, the same does not apply to O(KNL). In your example, KNL is nearly four orders of magnitude larger than N.**
> > > - **A7:** As you pointed out, we have revised the complexity analysis in the revised manuscript on page 6.
> > >
> > > - **Q8: You mentioned clarifying this in the NeurIPS rebuttal, but it was not addressed.**
> > > - **A8:** We have explained the "inverse sample difficulty" in the caption of Figure 1 in the revised manuscript on page 2, specifically,
> > >
> > >    - *Inverse sample difficulty: the reciprocal of sample difficulty.*

---

> ### Author Response · Authors · 2025-11-23
> **Looking forward to the reply**
>
> Dear Reviewer j34f,
>
> Thank you for your time and for the feedback provided earlier. Based on your comments, we have made substantial revisions to the manuscript, including:
>
>    - a comprehensive methodological comparison with EntAugment (Appendix F),
>    - additional comparisons with the published results of TrivialAugment (Appendix G),
>    - an updated and more rigorous efficiency analysis,
>    - and several clarifications of the main text.
>
> As the discussion period is coming to a close, we would greatly appreciate it if you could let us know whether there are any remaining concerns that we can further address. Your feedback at this stage is important for ensuring a fair and complete evaluation of the submission.
>
> Sincerely,
>
> The Authors

---

### Official Review · Reviewer_gLX1 · 2025-10-31

**Soundness:** 3
**Presentation:** 2
**Contribution:** 2
**Rating:** 4
**Confidence:** 3

**Summary:**

The paper introduces a sample-aware data augmentation strategy that dynamically adjusts the augmentation strength for each sample. This adjustment is based on estimating a sample's influence by projecting its gradient onto the model update direction. Specifically, the method applies stronger augmentation to samples exhibiting low variance and weaker augmentation to those considered unstable. This augmentation process is performed on-the-fly, and experimental results demonstrate its efficacy across various classification tasks.

**Strengths:**

- The core idea of dynamically adjusting augmentation strength based on gradient information is both logical and potentially powerful.
- The work includes a comprehensive set of experiments on classification tasks, covering various settings such as closed-set and open-set, and different k-shot scenarios.

**Weaknesses:**

- The paper does not sufficiently detail the computational overhead. Since the gradient must be calculated at every optimization step to determine the augmentation strength, a clear analysis of the computational complexity and the resulting wall-clock time overhead during training is necessary.
- The experimental results show varying degrees of advantage: a noticeable gap on Tiny-ImageNet, a moderate improvement on CIFAR-10, and almost similar test accuracy for ImageNet-1k (Table 7). Given the potential training time overhead, the marginal benefit on large-scale datasets like ImageNet-1k is not sufficiently advantageous to justify the added complexity.
- The method's performance appears to be highly sensitive to the choice of hyperparameters, specifically the window size and decay factor. Although the optimal window size shows a consistent decreasing trend, the optimal decay factor does not exhibit a clear tendency, which raises concerns regarding the robustness of the method. This suggests a user might need to perform an extensive grid search for the decay factor whenever the experimental setting (e.g., training data, classifier architecture) is changed.

**Questions:**

- Given the variety of augmentations (e.g., geometric, color), is the same decay factor applied universally across all categories of augmentation, or are different decay factors employed for different augmentation categories?
- Would the proposed sample-aware augmentation strategy offer benefits when applied to image generation tasks?

---

> ### Author Response · Authors · 2025-11-20
>
> Dear Reviewer gLX1,
>
> We sincerely thank you for the careful review and the insightful questions and comments. We appreciate your recognition of our work's strengths, particularly the powerful idea and strong results.
> For the comments and questions, we provide the point-by-point responses here:
>
> - **Q1: Since the gradient must be calculated at every optimization step to determine the augmentation strength, clear analysis of the computational complexity and the resulting wall-clock time overhead during training is necessary.**
> - **A1:** Thanks for your careful review and constructive suggestion. We would like to clarify that we adopt a first-order Taylor expansion to convert the gradient-projection term into a loss-difference formulation, which can be obtained directly from the forward pass. This avoids any additional gradient calculation beyond standard training, and thus, **the resulting computational overhead introduced by SADA is minimal.**
>
>   As suggested, we also analyze the actual wall-clock time on ImageNet-1k across various deep models. As shown in the table below, SADA incurs no noticeable additional training cost compared to standard training. We have included these results in Appendix E on page 15 in the revised manuscript.
>
> **Table B-1:** Wall-clock time (h) of the baseline and SADA on ImageNet-1k using a 4-A100-GPU server.
> ||ResNet-50| ViT-B| ViT-L|
> |- |- |- |- |
> |Baseline| 22.1|149.1 |363.2 |
> |SADA| 22.5|150.8| 366.4|
> ||+1.8%|+1.1%|+0.8%|
>
> - **Q2: Given the potential training time overhead, the marginal benefit on large-scale datasets like ImageNet-1k is not sufficiently advantageous to justify the added complexity.**
> - **A2:** Thanks for the comment. As clarified in our response to Q1, SADA introduces no meaningful training overhead.
> Regarding the benefit on ImageNet-1k, it is important to note that most strong baselines (e.g., Cutout, AA, FAA, DADA, TA, RA) improve by less than 0.8%. In contrast, SADA achieves 0.8-1.4% improvements over these methods, which constitutes a substantially larger gain. Therefore, SADA achieves the best trade-off between effectiveness and efficiency.
>
>    As a result, although ImageNet-1k leaves little headroom, SADA achieves top-tier performance with larger gains than most SOTA baselines with almost no additional training time overhead. The benefit is thus **NOT** marginal.
>
>
> - **Q3: The method's performance appears to be highly sensitive to the choice of hyperparameters, specifically the window size and decay factor.**
> - **A3:** Insightful comment. We would like to clarify that, the window length $L$ and decay factor $\beta$ are fixed to $L=10$ and $\beta=0.9$ across all datasets and architectures in our experiemnts, **without any dataset- or model-specific tuning.**
> Furthermore, our ablation study in Sec. 4.8 shows that small window sizes (e.g., L=10) and the decay factor exhibit highly stable performance. These findings demonstrate that the method **is not sensitive to these hyperparameters and that their default values work robustly across tasks**.
>
>   To fully address your comments, we have revised the discussion in Appendix I on page 16 to include practical parameter-setting suggestions. In practice, users can simply adopt a default configuration such as $L=10$ and $\beta=0.9$.
>
>
> - **Q4: Is the same decay factor applied universally across all categories of augmentation, or are different decay factors employed for different augmentation categories?**
> - **A4:** Yes. The same decay factor is **applied universally** across all categories of augmentation.
>
> - **Q5: Would the proposed sample-aware augmentation strategy offer benefits when applied to image generation tasks?**
> - **A5:** Thanks for the question. We acknowledge that our method is designed for image classification, e.g., the influence estimation in Eq. (6) relies explicitly on class-discriminative loss dynamics.
> While the general idea of adapting the augmentation strength based on sample-specific learning signals may conceptually extend to image generation, applying it to generative models would require substantial re-formulation due to their fundamentally different training objectives. Exploring sample-aware augmentation in generative tasks is, in our opinion, out of scope here, but of interest for our future work.
>
>   In response, we have updated the discussion section in *Appendix I* in the revised manuscript on page 16 to further discuss this point.

---

> ### Author Response · Authors · 2025-11-23
> **Looking forward to the reply**
>
> Dear Reviewer gLX1,
>
> Thanks so much again for the time and effort in our work. According to the comments and concerns, we have conducted the corresponding experiments and further discussed the related points. Additionally, according to your suggestions, we have included more evaluation results and discussions.
>
> As the discussion period is coming to a close, may I know if our rebuttal addresses the concerns? If there are further concerns or questions, please feel free to let us know.
>
> Thank you again for your thoughtful and constructive comments.
>
> Sincerely,
>
> The Authors

---

### Official Review · Reviewer_WfCB · 2025-10-31

**Soundness:** 3
**Presentation:** 3
**Contribution:** 3
**Rating:** 6
**Confidence:** 3

**Summary:**

The paper proposes SADA, a plug-and-play augmentation scheme that adapts per-sample augmentation strength on the fly using training-dynamics signals. At each step, the method first estimates a sample’s influence by projecting its gradient onto the accumulated model-update direction, and then measures stability as the temporal variance of this influence over a short window (EMA-smoothed). Stable (low-variance) samples receive stronger augmentations; unstable (high-variance) samples receive milder ones to preserve semantics. The influence is made efficient via a first-order loss-difference approximation. A bound is sketched that links SADA to reduced generalization complexity via a per-sample sensitivity term. Experiments on CIFAR-10/100, Tiny-ImageNet, ImageNet-1k, several fine-grained datasets, and long-tailed benchmarks report consistent gains and favorable accuracy-cost tradeoff.

**Strengths:**

1. The paper proposes an intuitive yet effective data augmentation approach that adapts per-sample augmentation strength on the fly. The methods takes sample variance into consideration via gradient projection, and avoids intense per-sample computation via a series of approximation, making it practical in a wide range of classification tasks.

2. The proposed method shows consistent performance improvement in extensive experimental settings. The method outperforms other methods on CIFAR-10/100, Tiny-ImageNet, and is competitive on ImageNet-1k. The method also achieves improvements on transfer-learning, fine-grained and long-tail datasets.

**Weaknesses:**

1. The proposed method introduces a principled approach for sample-aware augmentation. However, in each step a random augmentation operation is selected. Different image transformation process can impact the sample at different levels, which may interfere with the delicately designed augmentation strength.

2. The method introduced hyper parameters like window size and decay factor. Their values seem to be set based on experimental guidance, which probably need to be tuned individually for different tasks (standard, transfer or long-tail classification) and datasets.

**Questions:**

1. How does the method perform with controlled randomness for image transformation operations? Or is the random selection of operations an important factor in the approach?

2. How does the method affect the training dynamics? In particular, would the sample difficulty score in Figure 1 be distributed more evenly at the latter stage of training?

---

> ### Author Response · Authors · 2025-11-20
>
> Dear Reviewer WfCB,
>
> We sincerely thank you for the careful review and insightful comments/questions. We appreciate your recognition of our work’s strengths and provide point-by-point responses to address the comments raised.
>
> - **Q1/Q2: The impact of image transformations and the performance with controlled randomness in augmentation operations.**
> - **A1/A2:** Insightful comments. We would like to clarify that explicitly comparing the "impact level" of different augmentation operations is inherently ambiguous, e.g., it is not well-defined whether a rotation alters an image "more" than a translation or a color jitter.
> Instead of ranking heterogeneous operators by their semantic impact, our framework focuses on determining the strength of the applied transformation. This design offers two advantages: (1) it **simplifies the decision complexity**, and (2) it provides **more direct and fine-grained control** over how strongly each sample is augmented.
>
>   Moreover, to fully address your comments, we evaluate the effect of controlling randomness by restricting the number of candidate augmentation operations in our augmentation space (Table 8 on page 13).
>   By reducing the size of the augmentation space, the randomness in operator selection is correspondingly reduced. As shown in the table below, our method **remains highly stable across different levels of operation randomness**, with only a slight accuracy variation when fewer operations are available.
>
>   These results demonstrate that the superior performance of our method stems from the **adaptive adjustment of augmentation strengths**, rather than from the random selection of operations.
>
>   We have included these results in Appendix D on pages 14-15 in the revised manuscript.
>
> **Table A-1:** Performance under different numbers of augmentation operations in our augmentation space on CIFAR-100 using ResNet-50.
> |M|4|6|8|10|12|14 (Ours) |
> |- |- |- |- |- |- |- |
> |Acc. (%)|81.6|81.6|81.6|81.8|81.9|81.8|
>
>
> - **Q3: The method introduced hyperparameters like window size and decay factor. Their values seem to be set based on experimental guidance, which probably needs to be tuned individually for different tasks (standard, transfer, or long-tail classification) and datasets.**
> - **A3:** Thank you for your careful review. We would like to clarify that the window length $L$ and decay factor $\beta$ are **consistently fixed to $L=10$ and $\beta=0.9$** across all tasks (standard, transfer, or long-tail classification) and across all datasets, **without task-specific or dataset-specific tuning.**
> Moreover, our ablation study in Sec. 4.8 shows that smaller window sizes consistently yield strong performance, and the decay factor remains highly stable across a wide range of values. These findings indicate that our method is not sensitive to these hyperparameters and does not require careful tuning.
>
>   To further address your comments, we have revised the discussion in Appendix I on page 16 to include practical parameter-setting suggestions. In practice, users can simply adopt a default configuration such as $L=10$ and $\beta=0.9$, without any need for extensive hyperparameter search.
>
>
> - **Q4: How does the method affect the training dynamics? In particular, would the sample difficulty score in Figure 1 be distributed more evenly at the latter stage of training?**
> - **A4:** Thank you for your question. As shown in Fig. 1, we would like to clarify that the distribution of sample difficulty does not become even over time. Instead, as training progresses, the proportion of easier samples increases, while the proportion of difficult samples decreases. In the latter stage of training, most samples become easy for the model, reflecting its convergence on simpler patterns.
>
>    This evolution underscores the importance of **dynamic sample-aware augmentation**. Our method adjusts the augmentation strength according to this shifting difficulty distribution, helping prevent overfitting and improve overall generalization.

---

> ### Author Response · Authors · 2025-11-23
> **Looking forward to the reply**
>
> Dear Reviewer WfCB,
>
> Thank you once again for your time, effort, and constructive feedback on our work. Based on the comments and questions, we have conducted additional experiments, provided detailed discussions, and revised the manuscript to improve clarity. We have also included new evaluation results as suggested.
>
> As the discussion period is coming to a close, we would like to kindly confirm whether our responses adequately address your concerns. If there are further concerns or questions, please feel free to let us know.
>
> Thank you again for your thoughtful and valuable comments.
>
> Sincerely,
>
> The Authors

---

### Author Response · Authors · 2025-11-20
**General Response**

## General Description:

Dear Area Chairs and Reviewers,

We sincerely thank all Reviewers and Area Chairs for the time and effort dedicated to reviewing our paper. We appreciate the constructive feedback from all reviewers - WfCB (R1), gLX1 (R2), j34f (R3), which has helped us further improve the clarity and presentation of the manuscript. We also thank the reviewers for acknowledging the strengths of our work, including (1) a novel and intuitive formulation for dynamic augmentation (R1, R2), (2) a technically effective and practical design (R1, R2, R3), and (3) comprehensive experimental evaluations and strong results (R1, R2, R3). The raised concerns are mainly concentrated on (1) more experiments and analyses, and (2) further refinement to improve clarity.

**Additional Experiments and Analyses:**

In our response, we provide additional experimental results and analyses that address the comments, including:

1. Evaluation under different randomness in our augmentation operations (R1: Q1/Q2) (Table A-1).
2. Training costs (wall-clock time) analysis (R2: Q1) (Table B-1).
3. Comparison with the suggested works (R3: Q4) (Table C-1).

**All corresponding results have been incorporated into the revised manuscript.** Moreover, we have also included more discussions, clarified method details, and provided suggestions for parameter settings. These revisions address all reviewer comments in detail.

Thank you again for your insightful feedback and for helping us further refine our work.

Sincerely,

Authors of Submission 19309

---

### Meta-Review · Area_Chair_TFEC · 2025-12-26

**Summary:**

The reviewers’ concerns primarily focus on the sensitivity of the hyperparameters (window size and decay factor), training costs, and performance on ImageNet.

**Reviewer Concerns:**

While most concerns are addressed by the authors, the only issue is the performance on ImageNet. The proposed method does not achieve better results than MA, but at much lower training cost (27 vs. 50).

**Reviewer Scores:**

The reviewers WfCB and gLX1 may raise their scores after the discussion, since most of their concerns are addressed, except for the performance.

---

### Decision · Program_Chairs · 2026-01-26

Reject